

# Combining cloud radar and radar wind profiler for a value added estimate of vertical air motion and particle terminal velocity within clouds

Martin Radenz[1], Johannes Bühl[1], Volker Lehmann[2], Ulrich Görsdorf[2], and Ronny Leinweber[2]

[1]Leibniz Institute for Tropospheric Research (TROPOS), Leipzig, Germany
[2]Meteorologisches Observatorium Lindenberg/Richard-Aßmann-Observatorium, Deutscher Wetterdienst, Tauche, Germany

*Correspondence to:* Martin Radenz (radenz@tropos.de)

**Abstract.** Vertical-stare observations from a $482\,\mathrm{MHz}$ radar wind profiler and a $35\,\mathrm{GHz}$ cloud radar are combined on the level of individual Doppler spectra to measure vertical air motions in clear air, clouds and precipitation. For this purpose, a separation algorithm is proposed to remove the influence of falling particles from the wind profiler Doppler spectra and to calculate the terminal fall velocity of hydrometeors. The remaining error of both vertical air motion and terminal fall velocity is estimated

to be better than $0.1\,\mathrm{m\,s^{-1}}$ using numerical simulations. This combination of both instruments allows direct measurements of in-cloud vertical air velocity and particle terminal fall velocity by means of ground-based remote sensing. The possibility of providing a profile every $10\,\mathrm{s}$ with a height resolution of $< 100\,\mathrm{m}$ allows further insight into the process scale of in-cloud dynamics. The results of the separation algorithm are illustrated by two case studies, the first covering a deep frontal cloud and the second featuring a shallow mixed phase cloud.

*Copyright statement.*

## 1   Introduction

Clouds are a key component of the Earth's climate system (Bony et al., 2015). They drive the hydrological cycle (Ramanathan, 2001) and influence the radiative balance (Ramanathan et al., 1989). The radiative effect of clouds and the efficiency of precipitation formation depend strongly on the clouds' microphysical structure, i.e. number concentration, size, shape and phase of particles. Cloud microphysics is strongly controlled by vertical velocity, as up- and downdrafts control temperature und supersaturation of an air parcel (Shupe et al., 2008; Korolev, 2007; Morales and Nenes, 2010). Hence, cloud lifetime and the production rate of cloud condensate is sensitively coupled to vertical motion (Korolev and Isaac, 2003; Donner et al., 2016). However, only few observations of in-cloud vertical air motions are available, leaving a major driver of cloud microphysics unconsidered. The current lack of process level understanding leads to large uncertainties in numerical climate simulations and

weather prediction models (Williams and Tselioudis, 2007).





Cloudy air is a complex multiphase, multivelocity and multitemperature physical system and there is an ongoing scientific discussion regarding the correct equations for describing the motion of moist atmospheric air in the presence of condensation (Bannon, 2002; Wacker et al., 2006; Bott, 2008; Makarieva et al., 2013, 2017). In this paper we explicitly distinguish between the velocity of the homogeneous gaseous mixture of dry air and water vapor (hereafter called air motion for brevity) and the

velocity of liquid and solid water particles with respect to the surrounding air (hydrometeor terminal velocity). In the stationary case, the vertical velocity of hydrometeors, which can be observed by ground-based radars, is simply the superposition of air motion and terminal velocity.

Different radar based methods have been proposed for the retrieval of the vertical wind speeds in clouds, like the Mie notch retrieval (Kollias et al., 2002; Fang et al., 2017) and the dual frequency method (Williams, 2012). This paper presents a

dual frequency approach which combines Doppler spectra from a cloud radar and a radar wind profiler (RWP) to obtain this information. RWPs are the only remote sensing instruments which are sufficiently sensitive to scattering from the particle-free "clear air" (Van Zandt, 2000) and can be used to observe air motion directly. However, RWPs are not only sensitive to clear air scattering, but also to particle scattering from clouds and precipitation. This particle scattering can mask the clear air return and leads to the so-called Bragg-Rayleigh ambiguity (Knight and Miller, 1998). An algorithm is presented, which

tries to disentangle the contributions of both scattering mechanisms through a combination of Doppler spectra obtained by radars operating at two widely spaced frequencies. The simultaneous observation of air motion and particle velocity makes it furthermore possible to retrieve the particle terminal velocity.

The paper is structured as follows: First the theoretical background is provided in Sect. 2. Campaign and technical details of the instruments are presented in Sect. 3. Afterwards the algorithm is introduced and illustrated by two case studies. Artificially

generated spectra together with a Monte Carlo approach are used to evaluate the accuracy of the proposed algorithm and to provide an error estimate for each spectrum operationally. This evaluation is presented in Sect. 6 followed by a short discussion, a summary and the conclusions.

## 2   Theoretical background

The general form of the weather radar equation can be written as (Doviak and Zrnic, 1993)

$$P(\boldsymbol{r}_0) = \int_V I(\boldsymbol{r}_0, \boldsymbol{r}) \eta(\boldsymbol{r}) d^3 \boldsymbol{r} \tag{1}$$

where $\boldsymbol{r}_0$ is the center of a range bin and $\eta$ denotes the volume reflectivity. $I(\boldsymbol{r}_0, \boldsymbol{r})$ is the instrument weighting function. It depends on the antenna radiation pattern, especially the beamwidth and the characteristics of the pulse, most importantly pulse length. Its rather complex form can under certain assumptions be simplified into the calibration constant $C/r^2$ (detailed derivation in Doviak and Zrnic, 1993).

Two atmospheric scattering mechanisms are relevant for RWP, clear air scattering caused by inhomogeneities of the refractive index at a scale of half the radars wavelength (the Bragg scale) and scattering by hydrometeors. Clear air scattering, often



called Bragg scattering in short, can be used to assess vertical air motion without a particle proxy. While higher-order effects like fluxes of the local refractive index parameter may introduce differences between the observed Doppler velocity and true air velocity (Tatarskii and Muschinski, 2001; Muschinski, 2004; Muschinski et al., 2005; Muschinski and Sullivan, 2013), these effects are assumed to be mainly relevant in the convective boundary layer (Cheinet and Cumin, 2011). The volume reflectivity

of clear air $\eta_{\text{air}}$ can be related to the refractive index structure parameter by the Ottersten equation (Hardy et al., 1966; Ottersten, 1969; Fukao and Hamazu, 2014), if the Bragg scale lies within the inertial subrange of fully developed turbulence:

$$\eta_{\text{air}} = 0.38\lambda^{-1/3}C_n^2 \tag{2}$$

In the case of incoherent scattering, i.e. random distribution of the particles within the radar resolution volume, the volume reflectivity can be described with:

$$\eta_{\text{particle}} = \int \sigma(D)N(D)\,\mathrm{d}D = \frac{\pi^5|K|^2}{\lambda^4}Z. \tag{3}$$

where $\sigma(D)$ is the backscattering cross section, $N(D)$ the particle number distribution, $Z$ is the reflectivity factor and $|K|^2$ accounts for the refractive index of the particle.

Coherent radars are able to estimate the Doppler spectrum on the basis of the demodulated and pre-processed receiver data. The use of classical spectral estimators, like the discrete Fourier transform, is very well established in radar meteorology. The

obtained power spectrum describes the information content of the raw signal as a function of Doppler frequency. The Doppler spectrum contains the same amount of information as the raw data, if the assumption of a stationary random Gaussian process holds for the duration of the dwell (Note that this does not hold for some types of clutter echoes).

For RWP, clear air and particle scattering act simultaneously and independently, hence the processes are not correlated, which often leads to the appearance of two distinct spectral peaks in the Doppler spectrum (Gossard, 1988; Kollias et al., 2002;

Fukao and Hamazu, 2014). While this offers the principal option to retrieve the vertical wind speed even in the presence of precipitation, the applicability of this approach is rather limited since the two scattering peaks are often *entangled*, i.e. they can not be uniquely separated in the the Doppler spectrum as schematically illustrated in Fig. 1. Heuristic models for the Doppler spectrum have been proposed for the purpose of relating its properties to the physical parameters describing the scattering medium, however no comprehensive theory based on first principles is available. For particle scattering, the simple model for

the Doppler spectrum established by Wakasugi et al. (1986) has been widely used, see e.g. Williams (2016) or Fang et al. (2012). The particle signal is always shifted by the vertical air motion $v_{\text{air}}$, resulting in a observed velocity $v = v_{\text{air}} + v_t$ and it is furthermore broadened by turbulence within the resolution volume. No such model is yet available for the Doppler spectrum of a clear air scattering signal. The classical assumption that the clear air peak has a Gaussian shape was suggested for simplicity by Woodman (1985), however the author has also mentioned that deviations are not uncommon.

An unambiguous separation of both clear air and particle peaks in a single radars Doppler spectrum is sometimes possible, especially for VHF radars operating near $50\,\text{MHz}$, see e.g. Renggono et al. (2006), due to the strong increase of particle





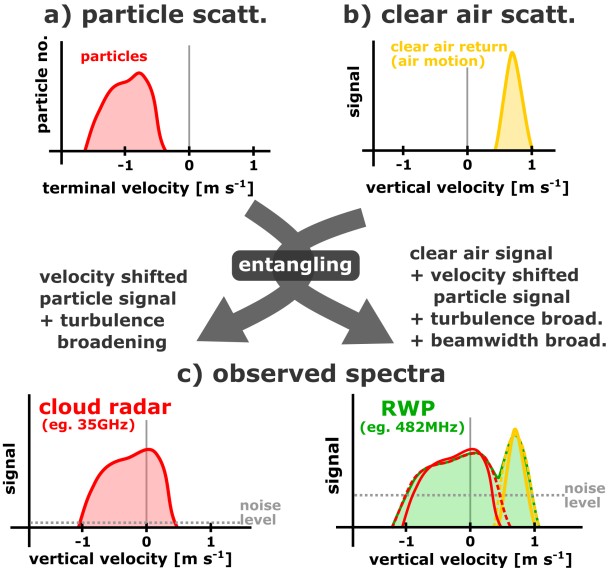

**Figure 1.** Scheme illustrating the contribution of particle and clear air scattering to the Doppler spectra of the radars. Scattering from particles (and their terminal fall velocity relative to the surrounding air) and the clear air return from the air motion are caused by completely different processes and have to be considered separately in an ideal case (a). Both processes are entangled, when observed by real instruments (b).

scattering reflectivity towards smaller wavelength, see Fig. 2. The dual frequency method attempts to resolve this entanglement by combining data from radars with sufficiently different wavelengths. The choice of frequencies is crucial to achieve a good separability. Gage et al. (1999) used a 1 and a 3 GHz radar to discriminate between clear air and particle scattering, yielding a frequency spacing factor of 3. Different methods were developed and further refined to separate both contributions (Williams

5  et al., 2000; Williams, 2002). This development lead to the approach of combining Doppler spectra from two radars operating at 50 and a 915 MHz, having a frequency spacing factor of 18 (Williams, 2012). The combination of a 482 MHz RWP and a 35 GHz cloud radar, as used in this study, provides a strongly increased frequency spacing factor of 73. The additional advantage of using a cloud radar operating at 35 GHz is that this instrument is de-facto not sensitive at all to clear air scattering, even for very high values of $C_n^2$. While a direct comparison of the theoretically calculated volume reflectivities (as in Fig 2)

10  would not immediately support this statement it needs to be considered that the Ottersten equation implicitly requires the existence of an inertial subrange at the Bragg scale. However, the Bragg scale of the cloud radar is only about 4 mm and therefore on the order of the Kolmogorov length microscale.

    A simple model for a single RWP Doppler spectrum $S(v)$ is given by Wakasugi et al. (1986)

$$S(v) = S_{\text{air}}(v) + S_{\text{particle}}(v) + N(v), \tag{4}$$





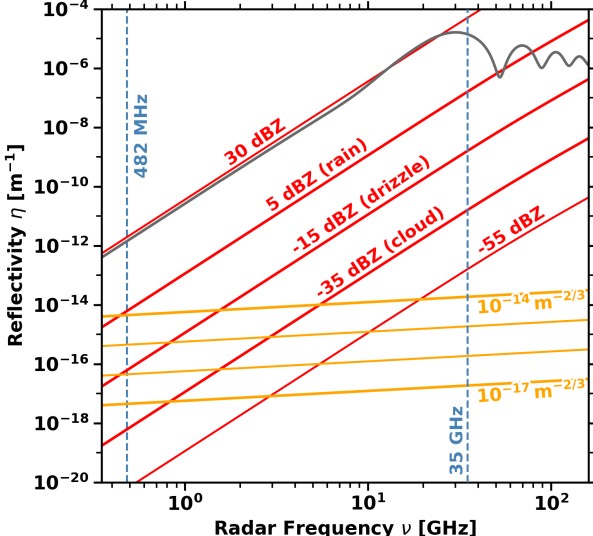

**Figure 2.** Reflectivity of different scattering processes. Red lines indicate typical values of hydrometeor reflectivity with the sensitivity threshold of the cloud radar at $-55\,\text{dBZ}$ (Görsdorf et al., 2015). Orange lines indicate the reflectivity for $C_n^2$ typically observed in the atmosphere (e.g. Clifford et al., 1994). The grey line shows the reflectivity of one liquid drop ($N = 1\,\text{m}^{-3}$) with a diameter of $d = 3\,\text{mm}$ as derived from Mie theory.

where $S_\text{air}(v)$ and $S_\text{particle}(v)$ are the spectra of clear air and particle scattering, respectively and $N(v)$ is the system noise. Following Atlas et al. (1973) and Kneifel et al. (2011), the received power can be decomposed through the Doppler spectrum $S(\boldsymbol{r}_0, v)$ as

$$P(\boldsymbol{r}_0) = \int S(\boldsymbol{r}_0, v)dv = \iint I(\boldsymbol{r}_0, \boldsymbol{r})\eta'(\boldsymbol{r}, v)\mathrm{d}^3\boldsymbol{r}dv \tag{5}$$

5    where $\eta'(\boldsymbol{r}, v)$ is the spectral reflectivity. For simultaneously acting clear air and particle scattering, the volume reflectivities are additive $\eta = \eta_\text{air} + \eta_\text{particle}$ (Rodgers et al., 1993), hence

$$\int S_{482}(\boldsymbol{r}_0, v)dv = \iint I_{482}(\boldsymbol{r}_0, \boldsymbol{r})[\eta'_\text{air}(\boldsymbol{r}, v) + \eta'_\text{particle}(\boldsymbol{r}, v)]d^3\boldsymbol{r}dv \tag{6}$$

$$\int S_{35}(\boldsymbol{r}_0, v)dv = \iint I_{35}(\boldsymbol{r}_0, \boldsymbol{r})\eta'_\text{particle}(\boldsymbol{r}, v)d^3\boldsymbol{r}dv \; . \tag{7}$$

This two relationships provide the basis for combination of the cloud radar and RWP Doppler spectra as it is described in

10    the following sections.





| | RWP | Cloud Radar |
|---|---|---|
| Type | LAP 16000 | MIRA 35 |
| Wavelength | 0.62 m | 8.5 mm |
| Beamwidth | 2.9° | 0.28° |
| Range Gate Length | 94 m | 30 m |
| Pulse Length | 1 µs | 200 ns |
| No. Coherent Integrations | 62 | 1 |
| Integration Time | 10 s | 10 s |
| No. Incoherent Averages | 4, 3 | 200 |
| Pulse Repetition Frequency | 12.2, 10.0 kHz | 5 kHz |
| Average Emitted Power | 200 W | 30 W |
| $N_{\mathrm{FFT}}$ | 512 | 256 |

**Table 1.** Configuration settings of the major instruments used in the COLRAWI campaigns (based on Bühl et al., 2015). The second number indicates the setting used for the RWP during IOPs.

## 3 Collocated observations with a 35 GHz cloud radar and a 482 MHz radar wind profiler

For this study we combine the data from the 35 GHz cloud radar (Görsdorf et al., 2015) and the powerful, narrow beamwidth 482 MHz RWP (e.g. Steinhagen et al., 1998; Lehmann et al., 2003; Böhme et al., 2004), both operated by the German Meteorological Service at Richard Aßmann Observatory in Lindenberg, Germany. Both radars were closely collocated to achieve
maximum overlap of the observation volumes. In the following, 'RWP' is used as an abbreviation for the 482 MHz radar and 'cloud radar' is used as a shorthand for the 35 GHz Doppler radar. It is worth mentioning that the proposed methods are not restricted to these frequencies.

The RWP is used with two configurations or operating modes. The first mode is used for intensive observation periods (IOPs), during which the RWP beam is pointing only into the vertical direction. During the second observation mode, 30 minutes of
vertical measurements are alternated with 30 minutes of Doppler beam swinging (DBS). This configuration was chosen to fulfill operational requirements and it also has the benefit that measurements of horizontal wind are available. The cloud radar is by design operating in vertical mode only. The technical parameters of cloud radar and RWP are given in Table 1. The measurements were performed between June and September 2015 in the framework of the COLRAWI campaign (Combined observations with lidar, radar and wind profiler; Bühl et al., 2015).
Figure 3 shows the measurements of both systems for an example case (17 June 2015; covered in detail in Sect. 5.1), where the Bragg-Rayleigh ambiguity becomes strikingly evident. A frontal system is approaching Lindenberg with the cloud base continuously decreasing from 6 km down to the ground as can be seen in the cloud radar reflectivity factor $Z_{35\,\mathrm{GHz}}$ (hereafter reflectivity). The RWP is able to measure the clear air signal, and hence the vertical velocity of clear air during the cloud free period (Fig. 3 d). As expected, the strongest clear air return is observed at the top of the atmospheric boundary layer (Fig. 3 c





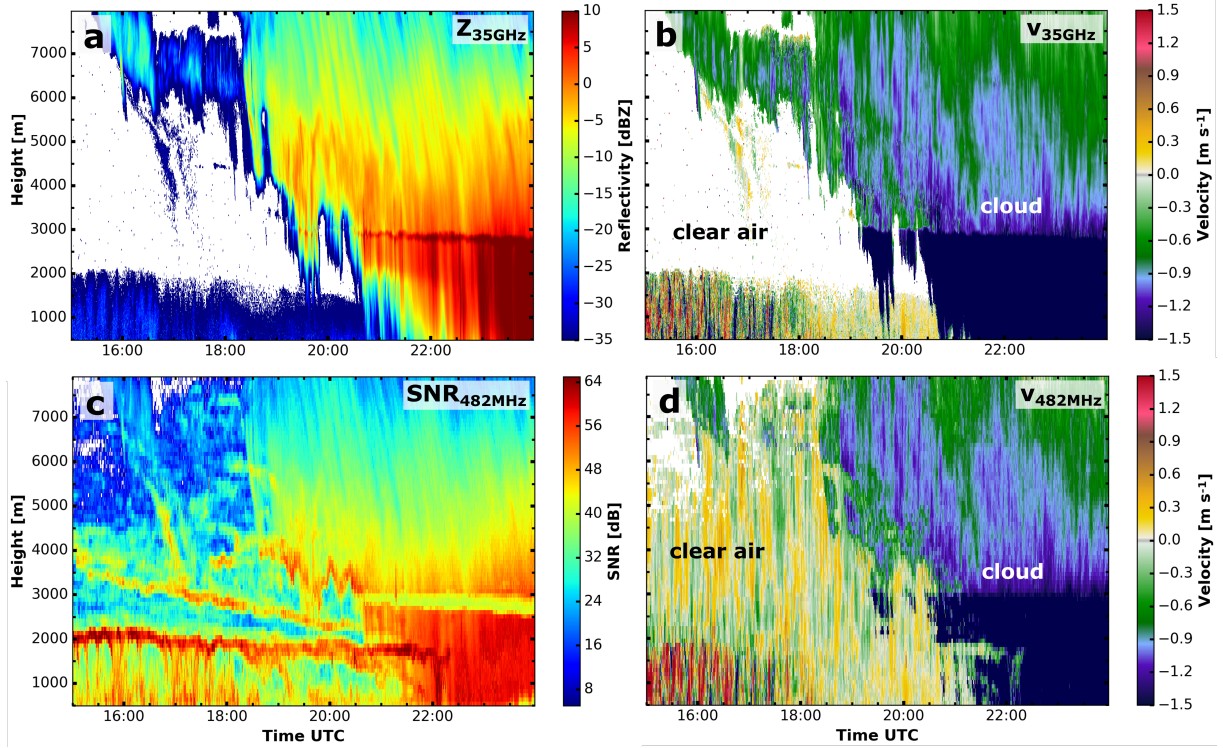

**Figure 3.** Cloud radar reflectivity (a), RWP SNR (c), cloud radar velocity (b) and RWP velocity (d) at the 17 June 2015 illustrating the Bragg-Rayleigh ambiguity. As soon as the particle return dominates (high reflectivity in the cloud radar), the clear air signal in the RWP is masked by the falling particles. This case is presented in more detail in Sect. 5.1.

at around $2.0\,\mathrm{km}$), but also at local maxima of refractive index gradients higher aloft. As soon as hydrometeors are present, the particle signal dominates and masks the air motion.

## 4   A synergistic algorithm to use cloud radar Doppler spectra for a suppression of particle echoes in RWP Doppler spectra

5   An overview of the proposed algorithm is given in Fig. 4. The Doppler spectra produced by the standard signal processing algorithm from the cloud radar (Görsdorf et al., 2015) and the RWP are used as input. In a first step, the effect of the differing sampling characteristics of both radars (especially beamwidth and pulse length) is accounted for: The signal peaks in the cloud radar spectra are artificially broadened and the range resolution is also coarsened. Furthermore, the power density in the RWP spectra are relatively calibrated based on the cloud radar data. The actual removal of the Bragg-Rayleigh ambiguity is

10   performed by suppressing the parts of the RWP spectra which are influenced by a particle signal. A peak finder is used with a moment estimation algorithm based on Gaussian fitting to estimate reflectivity, mean velocity and spectral width of the clear





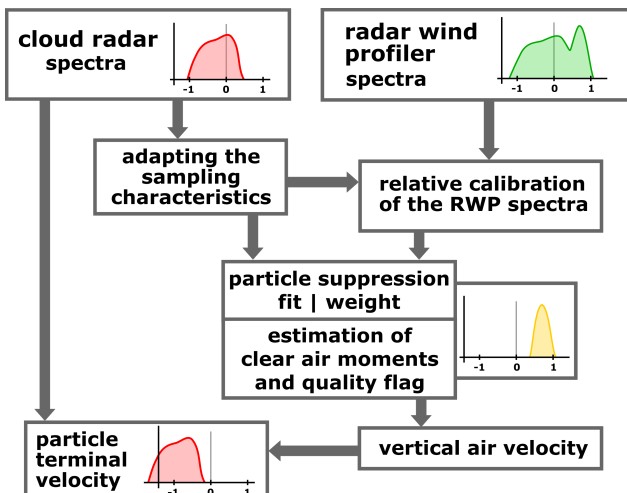

**Figure 4.** Scheme of the separation algorithm. The separation begins with the raw measurements from cloud radar and RWP as denoted in the top row. The separation between particle and clear air signal is performed after adaption of the different sampling characteristics and relative calibration. The products are the quality flagged vertical air velocity and the particle terminal velocity. Further description in the text.

air return. The particle terminal fall velocity is calculated by subtracting the RWP retrieved air velocity from the cloud radar derived Doppler velocity.

## 4.1 Adaption of the cloud radar spectra to match the sampling characteristics of the RWP

Both radars have significantly different instrument weighting functions, because of different antenna radiation patterns and
different pulse lengths (Table 1), which results in different resolution volumes. The pulse length is matched by summing the spectral reflectivities of the cloud radar spectra within a RWP range gate. A Gaussian window of $90\,\mathrm{m}$ width is used to account for the range weighting function of the RWP. The different beamwidth of the RWP is accounted by an artificial broadening of the cloud radar Doppler spectrum. A larger beamwidth is more susceptible to spectral broadening caused by the radial component of the horizontal wind at the edges of the beam. A full theoretical treatment is provided by Nastrom (1997), where
an analytical formula is provided, which is used to artificially broaden the cloud radar spectrum. This adapted cloud radar spectrum matches the sampling characteristics of the RWP and is referred to as $S_{35}(v)$ in the following and $\boldsymbol{r}_0$ is omitted for brevity. The profile of the horizontal wind required for this correction is taken from numerical weather prediction model data from the European Center for Medium-Range Weather Forecasts.

    Furthermore, both radars operate with different temporal sampling (also Table 1), which results in different frequency or
equivalently velocity resolutions of their Doppler spectra. The slightly different velocity resolution of the cloud radar is matched by linear interpolation, assuming uniformly distributed energy within each spectral bin. Additionally, the RWP spectrum is smoothed in a pre-processing step by convolution with a Gaussian window ($\sigma = 1\,\mathrm{bin}$) to reduce the variance of the estimated spectral power densities caused by the small number of coherent averages. It is referred to as $S_{482}(v)$.





## 4.2 Relative calibration of the RWP Doppler spectra

For a meaningful comparison of the RWP and cloud radar spectra, a relative calibration of of the RWP with the cloud radar as the reference is performed. Selecting a small velocity interval $[v_{\min}, v_{\max}]$, where both the Rayleigh approximation holds for the cloud radar and no clear air scattering contribution is present in the RWP, then equations 6 and 7 combine to

$$C_{482} = C_{35} \frac{\int\limits_{v_{\min}}^{v_{\max}} S_{482}(v)\mathrm{d}v}{\int\limits_{v_{\min}}^{v_{\max}} S_{35}(v)\mathrm{d}v} \tag{8}$$

with $C_{482}$ and $C_{35}$ the calibration constants of the RWP and the cloud radar, respectively. $S_{482}(v)$ and $S_{35}(v)$ are the for the sampling characteristics adapted Doppler spectra of RWP and cloud radar. This relationship holds, because the reflectivity factor $Z$ is independent of wavelength under the Rayleigh approximation. The calibration of the cloud radar is assumed to be correct with an accuracy of $1.3\,\mathrm{dB}$ (Görsdorf et al., 2015).

Above the boundary layer and in the absence of deep convection and strong gravity waves, the clear air signal of the vertical beam is always close to $0\,\mathrm{m\,s^{-1}}$ and therefore excluded if $v_{\max}$ is set to $-0.9\,\mathrm{m\,s^{-1}}$. The lower boundary of the velocity interval is intended to exclude non-Rayleigh scattering from large particles, which are characterized by a large terminal velocity (Heymsfield and Westbrook, 2010). Hence, $v_{\min}$ is set to $-3.0\,\mathrm{m\,s^{-1}}$, which corresponds to the terminal velocity of a liquid sphere with a diameter of $1\,\mathrm{mm}$.

For each Doppler spectrum the calibration is additionally checked by comparing the calibrated Doppler spectra $S_{482}(v)$ and $S_{35}(v)$ within the boundaries of the particle peak. If the difference between the spectral reflectivities is less than $2\,\mathrm{dB}$ in 4 bins around the maximum of the particle peak and less than $2\,\mathrm{dB}$ at its minimum the calibration is considered valid. Otherwise a correction factor is calculated by averaging $S_{482}(v) - S_{35}(v)$ in 4 bins around the maximum of the particle peak. The corrected calibration is applied, if this correction factor is less than $20\,\mathrm{dB}$ and the standard deviation less than $10\,\mathrm{dB}$. For larger correction factors, the calibration is flagged as unreliable.

This allows the automated estimation of the calibration constant within a wide range of atmospheric conditions and also a continuous monitoring of this relative calibration. Within the three months of the COLRAWI campaign 2015 the daily calibration constant has shown a standard deviation of less than $1\,\mathrm{dB}$ (Fig. 5).

## 4.3 Suppression of the particle scattering contribution in the RWP Doppler spectra and estimation of the clear air moments

### 4.3.1 Weighting function

A weighting function $\mathcal{P}_{\mathrm{air}}$ is used to suppress the particle influence in the RWP spectrum. It describes the relative contribution of clear air scattering to the whole spectral reflectivity

$$\mathcal{P}_{\mathrm{air}} = 1 - \frac{S_{35}(v)}{S_{482}(v)}. \tag{9}$$




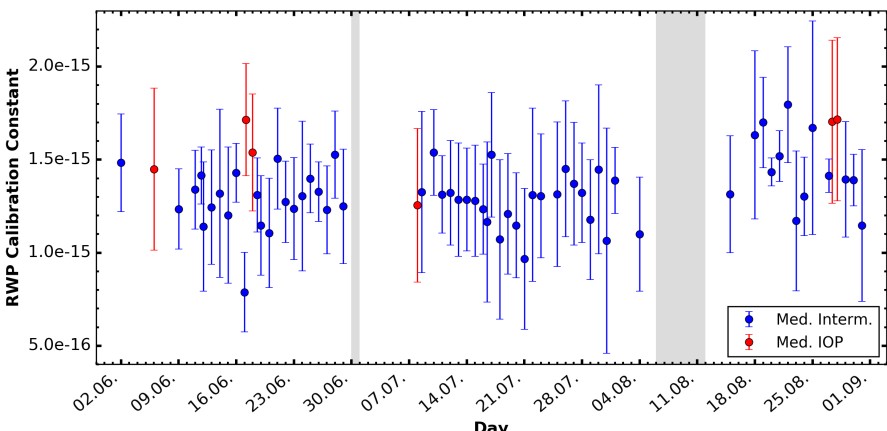

**Figure 5.** Daily median (dot) and the median absolute deviation (bar) of the RWP calibration for the whole campaign when appropriate conditions were present. The IOP settings are marked in red, the intermitting mode in blue. Measurement gaps due to maintenance are marked in grey. Note that the ordinate is linearly scaled.

The weighting function is defined to be equal to 1, if a bin in the Doppler spectrum is dominated by clear air scattering and 0 if particle scattering dominates. It is constructed as follows: The adapted cloud radar Doppler spectrum is cut off at the RWP noise level. The relative contribution is calculated from this cloud radar spectrum and the RWP spectrum using Eq. 9. Afterwards the weighting function is set to 1 at all bins where there is no cloud radar signal and is smoothed by a 5 bin wide

running mean to reduce noise. It is scaled with the inverse SNR of the cloud radar for all bins with a weight less than 0.5. This reduces the spectral reflectivity in the bins strongly dominated by particle return down to the noise level and provides a clear suppression of the particle contribution. The clear air reflectivity spectrum $S_{\mathrm{air}}(v)$ is then calculated by

$$S_{\mathrm{air}}(v) = \mathcal{P}_{\mathrm{scaled,air}}(v) \cdot S_{482}(v). \tag{10}$$

An example for a spectrum and the associated weighting function is shown in Fig. 6 (a, b). This weighting function approach

is similar to Williams (2012) with the difference that our weighting function is calculated for each bin individually without any cumulative distribution and the inverse SNR is used for scaling instead of a fixed $40\,\mathrm{dB}$ factor. The estimates for the first three moments (reflectivity, mean velocity and width) are the calculated using a standard moment estimator (e.g. Woodman, 1985; May and Strauch, 1989).

### 4.3.2 Peak fitting

A second method used to isolate the clear air contribution is fitting a Gaussian shaped function the part of the RWP spectrum which is not influenced by particle return. The peak fitting method is based on the assumption of a Gaussian shape of the



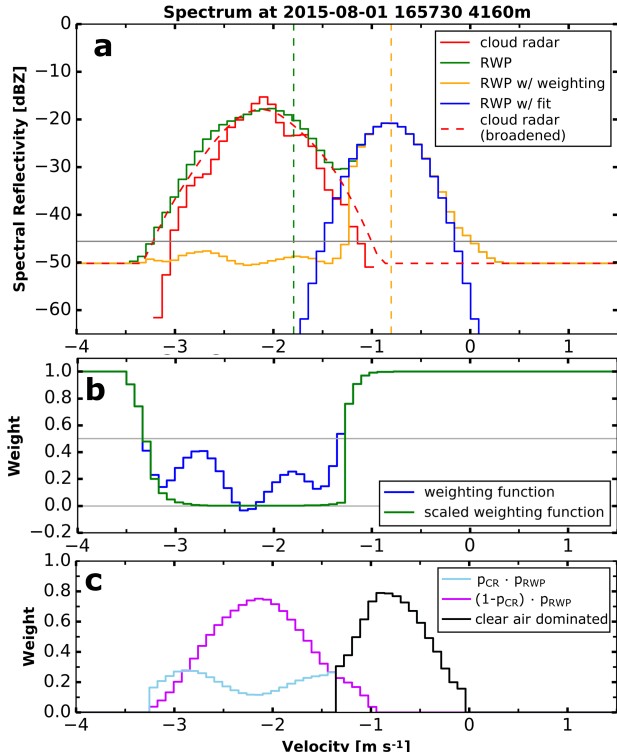

**Figure 6.** Doppler spectrum for 1 August 2015 16:57 UTC at 4160m (a), the associated weighting function (b) and the fuzzy membership function (c).

clear air peak (Woodman, 1985; Gossard et al., 1998). The fitting algorithm is constrained by a-priori information from the combined spectra together with long-term statistical properties of the clear air peak.

The bins without particle influence are identified a-priori by using a fuzzy membership like approach and a peak finding algorithm. A region in the spectrum is dominated by clear air return if

$$(1 - p_{\text{cloud radar}}) \cdot p_{\text{RWP}} > p_{\text{cloud radar}} \cdot p_{\text{RWP}}, \tag{11}$$

where $p$ is the spectral reflectivity of each instrument scaled to $[0, 1]$. All signal left of the cloud radar peak maximum is neglected. From this membership function (example in Fig. 6 c), the peak with the highest SNR is selected and its moments (reflectivity, mean velocity and width as calculated by a standard moment estimator) are used as a-priori information for the fitting algorithm. The Gaussian peak is then fitted to this part of the spectrum using a Trust Region Reflective algorithm (Branch et al., 1999). This fitting algorithm constrains the parameter space to physically reasonable values for the mean properties of the clear air peak in the absence of hydrometeors. The reflectivity $Z_{\text{air}}$ and the spectral width $\sigma_{\text{air}}$ are assumed to lie in the range between $-50$ and $10\,\text{dBZ}$ and $0.07$ and $0.45\,\text{ms}^{-1}$, respectively. The resulting fitting parameters are then identified as the moments of the clear air peak.





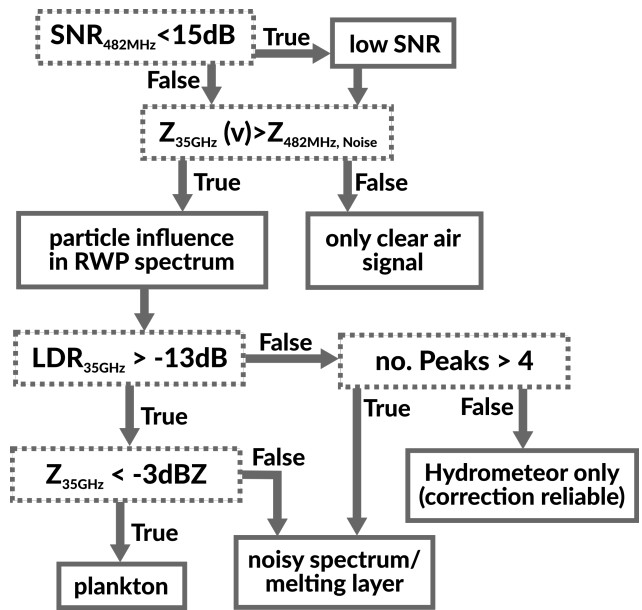

**Figure 7.** Flowchart of the quality flag decision logic.

## 4.4 Quality flag

A threshold based decision tree is used to determine the quality flag (Fig. 7). A RWP Doppler spectrum is considered to be particle influenced if the cloud radar spectrum contains bins where the cloud radar reflectivity is higher than the RWP noise level. Furthermore, spectra of the RWP with a SNR of less than $10\,\mathrm{dB}$ are flagged as low SNR. Spectra with a low SNR are not

5   necessarily less reliable, but during further processing, they have to be treated with care. In a next step the linear depolarization ratio (LDR) of the cloud radar is used to identify highly depolarizing targets (Reinking et al., 1997). The LDR threshold of $-13\,\mathrm{dB}$ is chosen to exclude all hydrometeors, which exhibit a smaller LDR (e.g. Matrosov, 1991), and is frequently employed in synergistic retrievals. The melting layer is characterized by high reflectivities combined with high LDR values in the cloud radar or an elevated noise level in the RWP. Due to the complex scattering processes within the melting layer (water coated

10   irregular spheres), a meaningful separation is not (yet) possible. Atmospheric plankton in the atmospheric boundary layer (like insects or pollen) is classified by high LDR and low reflectivity. For scattering by plankton the cloud radar frequently shows the same vertical air velocity as the RWP. The remaining thresholds are subjectively estimated based on visual inspection of the Doppler spectra.



## 5 Examples

### 5.1 Case Study 1: Frontal Clouds on 17 June 2015

In the following section, the separation algorithm is applied to the example case shown in Fig. 3. During the afternoon of the 17 June 2015, an occlusion with warm front character passed over Lindenberg. First high clouds appeared at about 16 UTC. The
cloud base slowly descended from above $6\,\mathrm{km}$ until liquid precipitation reached the ground at 23 UTC. The melting layer at around $2.8\,\mathrm{km}$ height is visible from approximately 19:30 onwards. Comparing Fig. 8f to Fig. 3d, the impact of the separation algorithm becomes visible. As shown in Fig. 8 b, high RWP reflectivity can be observed at strong gradients of temperature and/or humidity, both at the top of the atmospheric boundary layer at $2\,\mathrm{km}$, where reflectivities of up to $10\,\mathrm{dBZ}$ are visible as well as at the airmass boundary between $2.5\,\mathrm{km}$ and $6\,\mathrm{km}$. It becomes also visible how the precipitation alters the strong
reflectivity feature at the top of the boundary layer at about 22:20 UTC. The vertical air motion (Fig. 8f) reveals that the structure of successive up- and downdrafts sustains within the thinner parts of the cloud, especially before 19:30 UTC. Afterwards this pattern is less pronounced above the melting layer, whereas within the liquid precipitation the pattern of successive up- and downdrafts continues.

    The effect of the Bragg-Rayleigh ambiguity becomes also obvious in Fig. 9, which shows the frequency distribution of RWP
vertical velocities for the period shown in Fig. 8. The shapes of the distributions for clear air and raw RWP spectra differ significantly. The hydrometeors fall velocity causes a second mode at $-1.0\,\mathrm{m\,s^{-1}}$. After the separation, the distribution of vertical velocities within the cloud is rather similar to that of clear air velocities (in terms of mean and width).

### 5.2 Case Study 2: Mixed phase cloud on 1 August 2015

On 1 August 2015, a small-scale low pressure system over the eastern part of France initiated the development of high and
mid-level clouds in the southern part of Germany. During the day these clouds were advected towards Northeast. From 15:30 to 17:45 UTC a single layer mixed phase cloud was observed at Lindenberg. The RWP was operated in the intermitting mode on that day, meaning that 30 minutes of vertical stare are interrupted by 30 minutes of DBS. As the 18 UTC radiosonde ascent (Fig. 10 a and d) reveals a moist layer was present between $4$ and $6\,\mathrm{km}$ with a stable airmass aloft. Near cloud top (around $6\,\mathrm{km}$ height), the temperature was around $-17\,^{\circ}\mathrm{C}$ (Fig. 10a).

During the first part of the period shown in Fig. 10 the liquid water path (LWP) observed by a collocated microwave radiometer ranged between 70 and $120\,\mathrm{g\,m^{-2}}$, later it peaked at $190\,\mathrm{g\,m^{-2}}$. Beginning at 16:45 UTC ice production increased, forming a virga with a clear signature in the cloud radar reflectivity and vertical velocity below the liquid layer. The virga dissolves rather quickly in the dry layer below the cloud. In the radiosonde ascent (Fig. 10a) the relative humidity decreases from $100\%$ between $4.7\,\mathrm{km}$ and $5.5\,\mathrm{km}$ down to below $10\%$ at $3.8\,\mathrm{km}$. This dry layer is also visible in the RWP as gap in the
measurements.

    As the vertical air motion reveals in Fig. 11b, the dynamics are a key driver for this cloud. Regrettably only the intermittent mode observations are available for this case. But nevertheless, it is visible that the cloud gap at 16:30 UTC is caused by a downdraft. Before and after that downdraft the particles form or respectively grow when the liquid layer is lifted. It can also





**Figure 8.** Measurements and value added products during the evening of the 17 June 2015. Cloud radar reflectivity (a) and vertical velocity (c). RWP reflectivity (b), quality flag (d) and retrieved vertical air motion (f). The particle terminal velocity calculated from the air velocity and the cloud radar velocity (e). For the impact of the separation algorithm compare (a) and (f) to Figure 3. Areas flagged for quality reasons are colored white.

be seen, that a short delay exists between the maximum of the vertical velocity and the response (growth) of the particles (maximum of reflectivity and terminal velocity). This example also emphasizes the importance of the separation algorithm. The downward air motion at 16:35 UTC at $3.5\,\mathrm{km}$ and the virga 20 minutes later at $4.5\,\mathrm{km}$ could not be distinguished in the raw RWP measurement (Fig. 11a). It may also be possible, that the downdraft at 16:55 UTC was amplified by evaporation cooling.

5   Collocated observations of a vertical staring Doppler lidar (not shown) augments these findings by observing the updraft also in the liquid layer at cloud top.



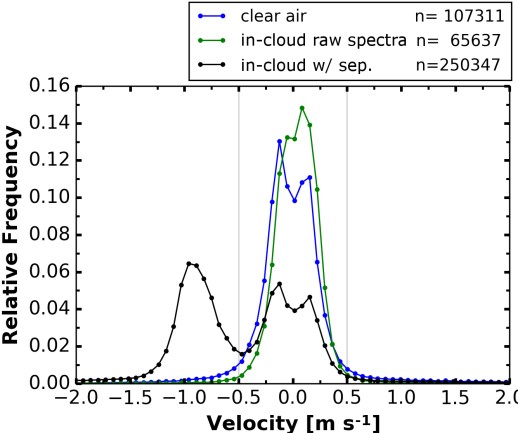

**Figure 9.** Histogram of RWP vertical velocities for the evening of the 17 June (same period as shown in figure 8). $n$ denotes the number of spectra, that contributed to each class. The minimum at $0\,\mathrm{m\,s^{-1}}$ in the clear air and in-cloud raw spectra is caused by the stationary clutter filtering procedure in the original RWP signal processing.

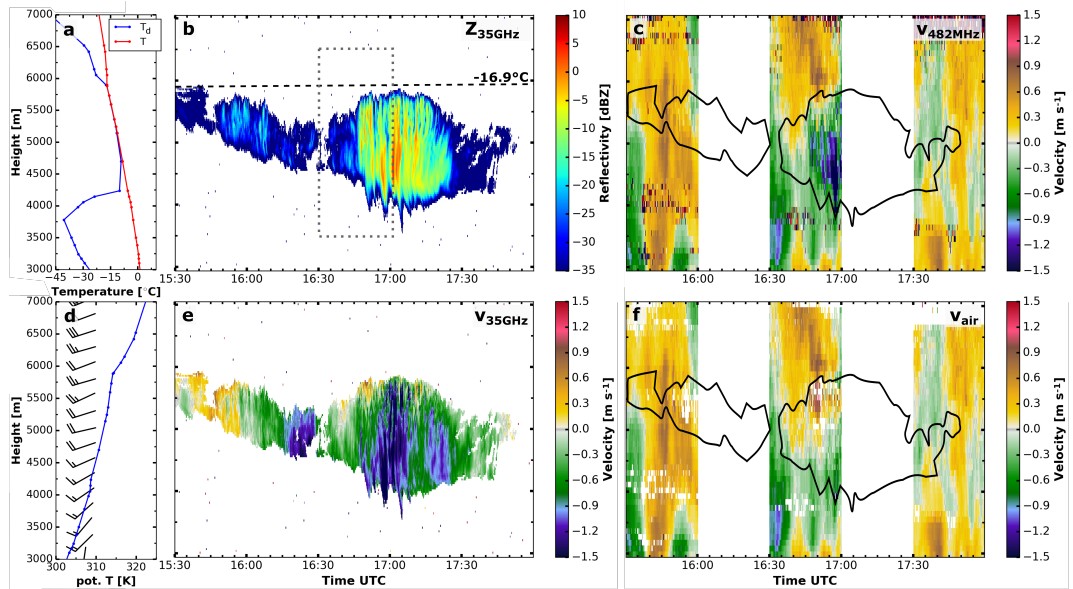

**Figure 10.** Mixed phase layered cloud in the afternoon of 1 August 2015. Temperature $T$ and dewpoint temperature $T_d$ (a) as well as potential temperature and wind profile from the 18 UTC radiosonde ascend. Cloud radar reflectivity (b), vertical velocity retrieved with the standard RWP signal processing (c), cloud radar vertical velocity (e) and the retrieved vertical air motion (f). Values exceeding the velocity color scale are shown in dark blue and red, respectively.





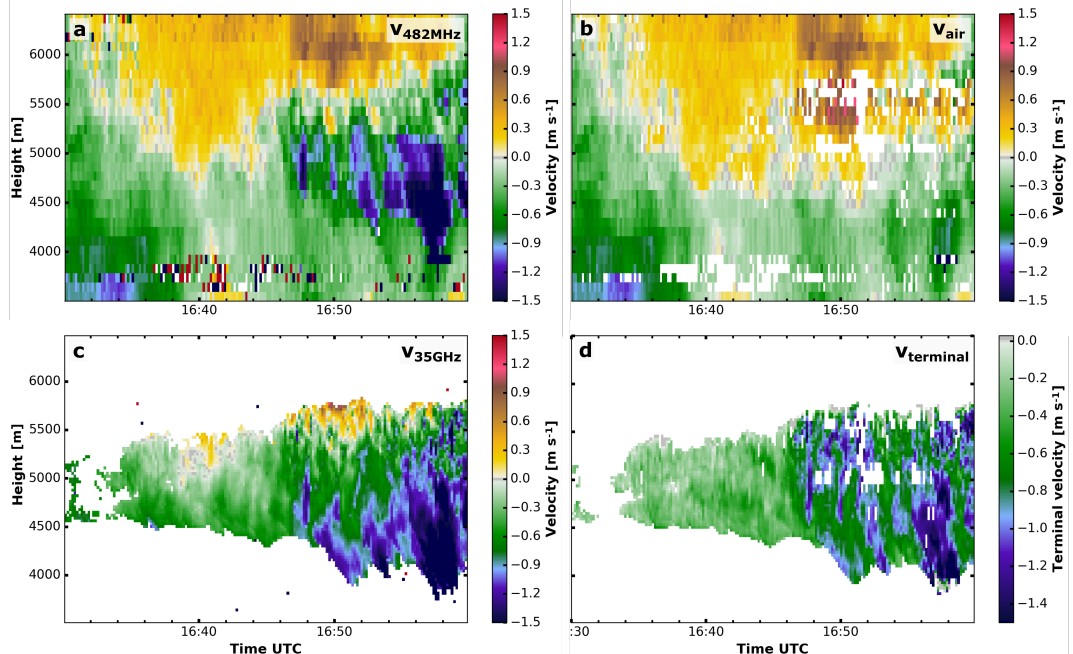

**Figure 11.** Zoom into the mixed phase layered cloud between 16:30 and 17:00 UTC. Vertical velocity retrieved with the standard RWP signal processing (a), retrieved vertical air motion (b), vertical velocity of the cloud radar (c) and the terminal velocity (d). Values exceeding the velocity color scale are shown in dark blue and red, respectively.

Having an estimate for in-cloud vertical air velocity, the terminal velocity of the particles can be calculated (Fig. 11d). For a first investigation, the mean velocity of the cloud radar peak was used, but the Doppler spectra could also be used. When the updraft strengthens at 16:45, the terminal velocity increases as the particles grow. The particles evaporate rather quickly when reaching the dry layer at $4\,\mathrm{km}$.

## 6   Evaluation of the separation algorithm

The accuracy of the separation algorithm is estimated with a Monte Carlo approach. Doppler spectra of cloud radar and RWP are generated numerically by composing a particle and a clear air peak

$$S_{482}\left(v\right) = P_{\mathrm{Gauss}}\left(Z_{\mathrm{particle}}, \overline{v}_{\mathrm{particle}}, \sigma_{\mathrm{particle}}\right) + P_{\mathrm{Gauss}}\left(Z_{\mathrm{air}}, \overline{v}_{\mathrm{air}}, \sigma_{\mathrm{air}}\right) \qquad (12)$$

$$S_{35}\left(v\right) = P_{\mathrm{Gauss}}\left(Z_{\mathrm{particle}}, \overline{v}_{\mathrm{particle}}, \sigma_{\mathrm{particle}}\right) \qquad (13)$$

with a Gaussian shaped peak $P_{\mathrm{Gauss}}$ having the first three moments $Z$, $\overline{v}$ and $\sigma$ These spectra are used as an input for the separation algorithm. The output of the algorithm is then compared with the input parameters of the synthetic Doppler spectra.

A single iteration in this Monte Carlo simulations consists of the following steps: At first, the input parameters (reflectivity, mean velocity and spectrum width) for both peaks are randomly chosen. The frequency distributions for the clear air peak is





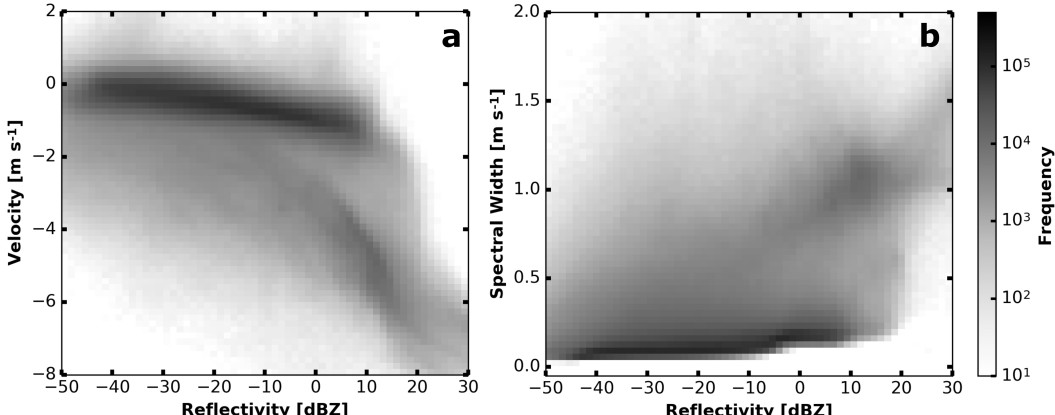

**Figure 12.** Dependency of the vertical velocity (a) and the width (b) on the reflectivity as observed by the cloud radar during COLRAWI. Clearly visible are the two clusters formed by cloud particles (low reflectivity and slowly falling) and precipitation (high reflectivity and faster falling).

| Parameter | Range |
|---|---|
| $Z_{air}$ | $[-35, 0]\,\mathrm{dBZ}$ |
| $\overline{v}_{air}$ | $[-1.0, 1.0]\,\mathrm{ms}^{-1}$ |
| $\sigma_{air}$ | $[0.1, 0.3]\,\mathrm{ms}^{-1}$ |

**Table 2.** Input settings for the Monte Carlo simulation. The values are based on the clear air properties of the RWP peak. Square brackets denote the range from which the random numbers are drawn.

derived from RWP observations within clear air. The parameters of the particle peak are closely coupled to each other and the parameters cannot be drawn from independent random distributions. Large reflectivity values are more common at higher fall velocities and spectral widths (Fig. 12). All cloud radar observations during the COLRAWI campaign are used to assemble a three-dimensional histogram. The randomly chosen reflectivity is based on the whole frequency distribution. Then a slice

5  through the histogram at this reflectivity is used to obtain the frequency distribution for which the velocity is drawn. The spectral width is chosen accordingly.

From these set of parameters, the synthetic Doppler spectra are calculated (Eq. 12 and 13). For the cloud radar spectrum only the particle peak is used, whereas for the RWP particle and clear air peak are added. Afterwards the noise floor and optionally multiplicative noise are added. These two synthetic Doppler spectra are in a next step used as input for the separation algorithm

10  described in Section 4. The randomly drawn input parameters and the output from the algorithm are stored for each step. If the separation algorithm fails to reveal the clear air peak, this is also stored.

By running multiple Monte Carlo steps, all combinations of input parameters are covered. Here, for each method (weighting function and peak fitting) 150000 Monte Carlo steps were used and the error of the vertical air velocity estimate of the algorithm





is calculated

$$v_{\mathrm{err}} = \overline{v}_{\mathrm{air,input}} - \overline{v}_{\mathrm{air,\ corr}}. \tag{14}$$

where $v_{\mathrm{air,input}}$ is the clear air velocity used as input and $v_{\mathrm{air,\ corr}}$ is the result of the separation process. If the obtained vertical velocity is larger than the actual one, $v_{\mathrm{err}}$ becomes negative, indicating an upward bias of the algorithm. If $v_{\mathrm{err}}$ is positive, there

is a downward bias. This error decreases if both contributions can be better distinguished in the spectrum. As a measure for the distance of the peaks we define the peak separation $\mathcal{S}$

$$\mathcal{S} = \frac{|\overline{v}_{\mathrm{air}} - \overline{v}_{\mathrm{particle}}|}{\sigma_{\mathrm{air}} + \sigma_{\mathrm{particle}}} \tag{15}$$

where $\overline{v}$ and $\sigma$ are the first and second moment, respectively, of the Doppler spectrum. For example, a particle peak at $\overline{v}_{\mathrm{particle}} = -1.5\,\mathrm{ms}^{-1}$ with $\sigma_{\mathrm{particle}} = 0.8\,\mathrm{ms}^{-1}$ and a clear air peak at $\overline{v}\mathrm{air} = +0.5\,\mathrm{ms}^{-1}$ with $\sigma_{\mathrm{air}} = 0.3\,\mathrm{ms}^{-1}$ give a peak sepa-

ration of 1.8. The spectral contrast $\mathcal{C}$ at $\overline{v}_{\mathrm{air}}$ is used as a second measure for peak distinguishability

$$\mathcal{C} = S_{\mathrm{air}}\left(\overline{v}_{\mathrm{air}}\right) - S_{\mathrm{particle}}\left(\overline{v}_{\mathrm{air}}\right). \tag{16}$$

The spectral contrast falls back to the RWP SNR, when the reflectivity of the particle signal is below the noise level at $\overline{v}_{\mathrm{air}}$. Fig. 13 a shows how the error depends on $\mathcal{S}$ and $\mathcal{C}$ for the particle influenced measurement. As it becomes clear from Fig. 13 b and c the error reduces with increasing $\mathcal{S}$ and $\mathcal{C}$. For peak separations $\mathcal{S}$ above 2 and spectral contrasts $\mathcal{C}$ above $15\,\mathrm{dB}$

the possible errors of both separation methods are negligible. The peak fitting approach performs slightly better if both, $\mathcal{S}$ and $\mathcal{C}$ are small, but at the cost of larger errors for small $\mathcal{C}$ and $\mathcal{S}$ above 1.8. Altogether the bias for the weighting function is on average $-0.023\,\mathrm{m\,s}^{-1}$ (median $-0.005\,\mathrm{m\,s}^{-1}$, interdecile range $0.072\,\mathrm{m\,s}^{-1}$) and for the peak fitting $+0.003\,\mathrm{m\,s}^{-1}$ ($0.001\,\mathrm{m\,s}^{-1}$, $0.058\,\mathrm{m\,s}^{-1}$), respectively.

A detection rate can be calculated from the portion of spectra where the separation was successful. The weighting function

approach is able to separate $86\%$ of all synthetically generated spectra and performs slightly better than the peak fitting approach ($80\%$). As evident from Fig. 14, the lowest detection rates occur at low peak separations $\mathcal{S}$ and spectral contrasts $\mathcal{C}$. Especially for peak separations around 1 the weighting function is more robust compared to the fitting approach.

Additionally to an estimate of the mean (statistical) error, a single error estimate for each spectrum is of great advantage. The measurements during the first case study (Sec. 5.1) for a error estimate for each spectrum. The observed Doppler spectra

from cloud radar and RWP are used to calculate $\mathcal{S}$ and $\mathcal{C}$. The error is then taken from the binned results from the Monte Carlo simulation (Fig. 13 b or c). The frontal cloud observed on 17 June 2015 (Sect. 5.1) is used to illustrate the error estimate for each spectrum (Fig. 15). Within liquid precipitation (below $3\,\mathrm{km}$), the error is negligibly small as clear air and particle contribution are clearly distinguishable in the spectrum. Above the melting layer the error is larger because of low terminal velocities, with the particle peak being very close to the clear air peak. The error of the weighting function is mostly negative,

indicating a slight upward bias. Contrarily the errors of the peak fitting method are mostly positive, meaning a slight downward bias.



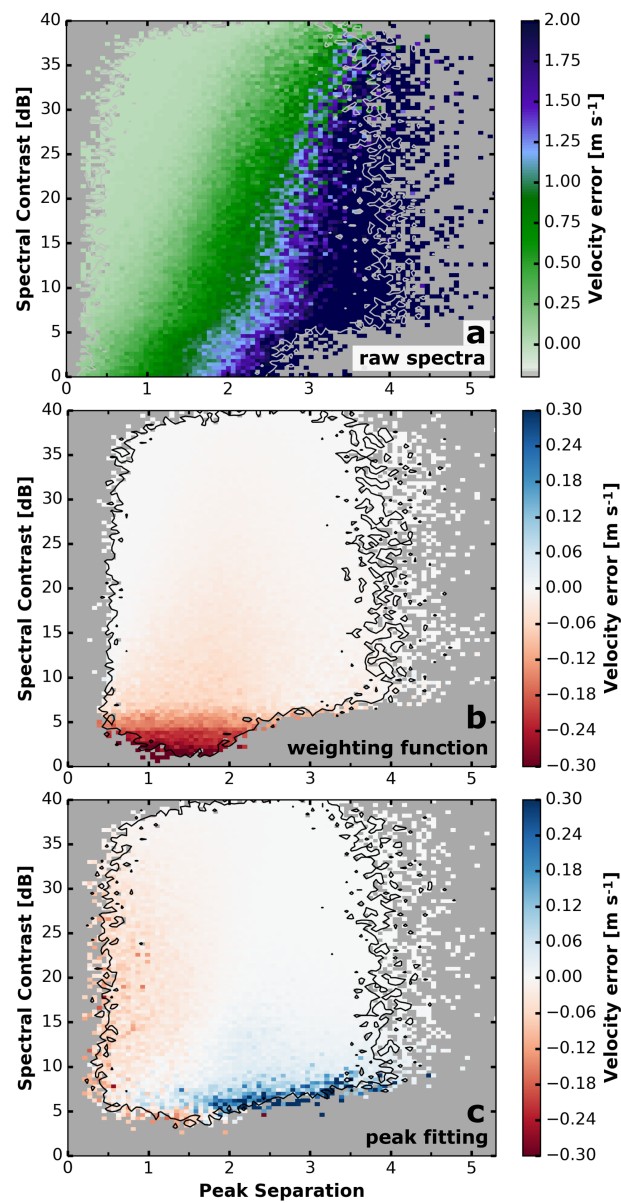

**Figure 13.** Velocity error before the separation (a), with the weighting function (b) and peak fitting (c). Shown is the bin mean for all Monte Carlo spectra within the respective bin. The black outline marks bins containing more than 7 values. Bins without any simulations are marked in gray.



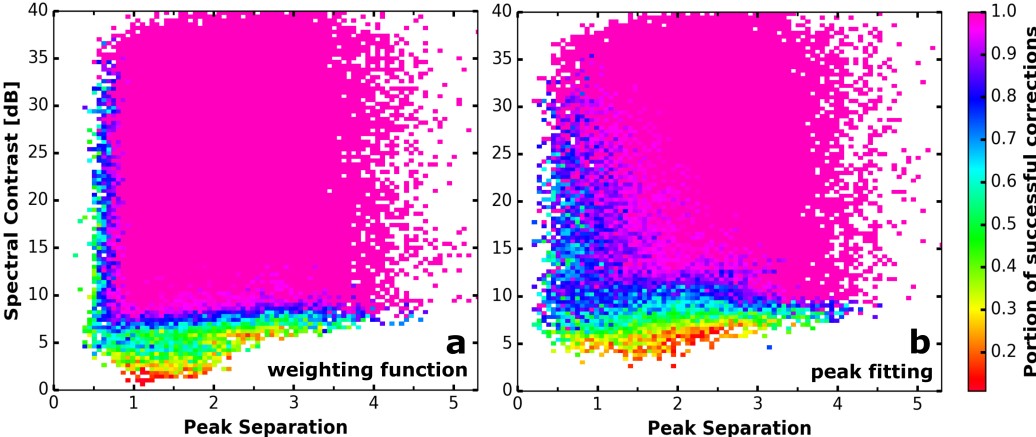

**Figure 14.** Portion of successfully separated spectra depending on the peak separation and contrast for the weighting function (a) and peak fitting (b)

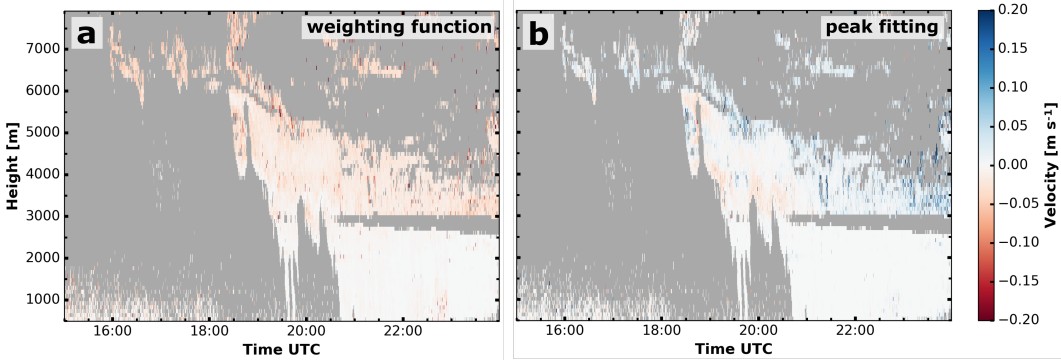

**Figure 15.** Error estimate for the weighting function (a) and the peak fitting (b) methods during the period covered in case study 1 (Sect. 5.1). Areas where no error estimate is possible are marked in gray.

## 7 Discussion

The Discussion will concentrate on three issues. The calibration is assessed and compared to previous work. Secondly the selection of frequencies is shortly discussed and the weighting function approach is also compared to prior work. Finally the error as estimated by the Monte Carlo simulation is shortly reviewed.

The accuracy of the relative calibration depends on the calibration of the cloud radar. According to Görsdorf et al. (2015) the internal budget calibration is accurate to $1.3\,\mathrm{dB}$. Neglecting all issues in the cloud radar calibration and including the variability of the RWP calibration constant (Sect. 4.2), the uncertainty in the RWP calibration is roughly $2\,\mathrm{dB}$. Orr and Martner (1996) applied a similar relative calibration approach as used in this study by calculating the reflectivity of the full spectrum within light rain events. The approach used in here has two advantages. Firstly it is not required to manually select of the events





where calibration is possible. Secondly non-Rayleigh contributions are excluded, which would otherwise mix into the SNR of the RWP and obscure the true calibration constant. The check of the calibration constant mitigates effects that decrease the observed reflectivity, like partial beam filling. Furthermore the correction factor may be used as a first assessment of attenuation due to liquid water.

As stated above, the choice of the frequencies governs the whole separation process. In contrast to a lower frequency radar, the 35 GHz system is completely insensitive to clear air scattering. Hence, its Doppler spectrum can be regarded as reference for the particle influence. A large frequency spacing factor of 73 supports an even stronger discrimination of clear air and particle signal.

Compared to Williams (2012), the separation scheme had to be modified in several aspects. The beamwidth of the cloud
radar is rather small ($0.28°$, Table 1), hence the beamwidth broadening effect had to be included (see Sect. 4.1). During construction of the weighting function, the fixed scaling factor of $40\,\mathrm{dB}$ was replaced by the inverse SNR, which provides a better suppression of the particle contribution. With the instruments used in this study, the resolution in terms of velocity ($< 0.1\,\mathrm{m\,s^{-1}}$), height ($< 100\,\mathrm{m}$) and time ($10\,\mathrm{s}$) is considerably improved compared to prior work (e.g., Williams, 2012), making the scale of cloud processes accessible.

The Monte Carlo approach can only provide a first estimate of the possible error. For single Doppler spectra the error in velocity according to the Monte Carlo simulation can be up to $\pm0.3\,\mathrm{m\,s^{-1}}$, whereas on average this error is much smaller. These large errors are typically caused by spectra where scattering from clear air and particles is hardly separable. The prediction of the error based on parameters computed from the Doppler spectrum allows to get an error estimate for quasi-instantaneous values of the vertical air velocity. This offers the possibility to include the error estimate in all following analysis steps, which
is an improvement compared to prior work.

## 8 Summary, Conclusions and Outlook

An synergistic algorithm based on a combination of Doppler spectra of a RWP and a 35 GHz cloud radar was developed with the goal of resolving the Bragg-Rayleigh ambiguity, which can mask the vertical air motion when particles are present. It was evaluated by a Monte Carlo approach using synthetically generated Doppler spectra. The bias in the vertical air velocity
estimate for both methods is close to $0\,\mathrm{m\,s^{-1}}$ with a interdecile range of below $0.1\,\mathrm{m\,s^{-1}}$. The results of the Monte Carlo simulations are used to provide an error estimate for single Doppler spectra. To automate the algorithm a continuous relative calibration procedure and a quality control flag were also included. The relative calibration proved to be quite stable over the three months of observations available so far.

The application of the separation algorithm for vertical air velocity estimate within clouds was shown for two case studies.
They illustrate that the algorithm can be applied to real measurements under various atmospheric conditions, offering a deeper insight into the formation and evolution of clouds.

The Cloudnet retrieval (Illingworth et al., 2007) provides a proven synergistic method for model evaluation (e.g., Morcrette et al., 2012; Neggers et al., 2012) and the long term quality controlled dataset makes also detailed cloud microphysics studies



possible (Bühl et al., 2016). However, continuous information on vertical air motion is not yet available in Cloudnet, which means that a major constraining factor of cloud microphysics is disregarded. The presented combination of a RWP and a cloud radar together with the separation algorithm is able close this gap and can provide a dataset for further studies on aerosol-cloud-dynamics interaction and model evaluation.

Besides the air motion, the calibrated RWP can provide additional information. Being less susceptible to attenuation, quantitative measurements of the reflectivity are possible under nearly all conceivable weather conditions. For long term measurements the combination of a 30 minute DBS based wind measurement mode followed by a 30 minutes high resolution vertical wind measurement mode turned out to be the best compromise to obtain a maximum of information. By combining the intermitting mode with standard Cloudnet methodology long term observations of vertical air motion on the scale of clouds become

possible. Such a dataset would allow for model evaluation and cloud-dynamics-interaction studies over statistically significant time periods.

*Code and data availability.*   The processing software 'spectra mole' is available at github (https://github.com/martin-rdz/spectra_mole). The raw and processed data is available from the corresponding author on request.

*Competing interests.*   The authors declare that they have no conflict of interest.

*Acknowledgements.*   The research leading to these results has received funding from the European Union Seventh Framework Programme (FP7/2007-2013) under grant agreement numbers 262254 (ACTRIS) and 603445 (BACCHUS) and from the HD(CP)$^2$ project (FKZ 01LK1209C and 01LK1212C) of the German Ministry for Education and Research.





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
