# Peer review of "Combining cloud radar and radar wind profiler for a value added estimate of vertical air motion and particle terminal velocity within clouds"

_Atmospheric Measurement Techniques, 2018_

## Referee Comment (RC1) · Anonymous Referee #1 · 10 Jul 2018

This manuscript describes how to estimate vertical air motion using two vertically pointing radars operating at different frequencies positioned next to each other. The longer wavelength radar (frequency of 482 MHz) can detect both Bragg scattering from turbulence refractive gradients and Rayleigh scattering from hydrometeors. The shorter wavelength radar (frequency of 35 GHz) is sensitive to Rayleigh and non-Rayleigh scattering from hydrometeors. By examining the Doppler velocity spectra from both radars, the Bragg scattering signal can be isolated in the 482 MHz radar spectra and used to estimate the vertical air motion during precipitation events.

General Comments

The analysis presented in this manuscript is appropriate for Atmospheric Measurement Techniques. After a couple suggested changes, this manuscript should be ready for publication.

Specific Comments

1. Equations (1), (5), (6), and (7). The term dˆ3 is not needed in these equations and is not defined in the text. I expect that "d" is raindrop diameter, but "D" is defined as the raindrop diameter in the text. Also, the variable eta(r) denoting the volume reflectivity includes the influence of raindrop diameter. Thus, raindrop diameter does not need to be explicitly stated in the equations.

2. Page 3, line 10. For completeness, the backscattered cross section sigma(D) should be a function of radar operating frequency (or wavelength). Since the backscattering cross section at 35 GHz includes both Rayleigh scattering for small drops and non-Rayleigh scattering from larger drops, this should be reflected in the variable of sigma(D).

3. Page 3, line 10, equation (1) and also page 9, lines 1-14. Since the calibration is using small raindrops that are in the Rayleigh regime, it would make sense to include after equation (1) the Rayleigh and non-Rayleigh dependence in the 35 GHz backscattering cross-section (see comment #2 above). While this is discussed in the Discussion section (page 21, line 1), Rayelgih and non-Rayleigh scattering should be mentioned near the text developing equation (1).

4. Page 9, lines 1-14. Is there any attenuation correction performed while implementing the calibration procedure? Even if there is not an attenuation correction performed, please include in the manuscript the need to account for attenuation at 35 GHz while performing the reflectivity estimation. Also, is this calibration procedure only performed at low altitudes, when the attenuation will be smaller than at further ranges?

5. Page 4, lines 1-12, and Fig. 2. The diagram shown in Fig. 2 reminds me of a figure

from Ralph (J. Atmos. Oceanic Technol. 257-267, 1995) showing the sensitivities of Bragg and Rayleigh scattering. But, the Ralph figure only goes up to a frequency of 10 GHz.

6. Page 8, line 1. Please clarify the expression, "is calculated by subtracting the RWP retrieved air velocity from the cloud radar derived Doppler velocity". This "subtraction" could mean the subtraction of power within Doppler velocity spectrum, or could mean the subtraction of velocity moments between different radars. Also, the "subtraction" could mean the shift in Doppler velocity spectrum before estimating the moments.

I thought the rest of the manuscript read very well with good descriptions of the algorithm and the applications.

---

## Referee Comment (RC2) · Anonymous Referee #2 · 10 Aug 2018

The paper presents a method to separate cloud/precipitation radar returns from clear air returns using a 35 GHz cloud radar and a 482 MHz radar wind profiler (RWP). Previous work by Williams and others has shown the utility of using two RWP (UHF and VHF) to separate air motions and hydrometeor terminal velocity. The use of a 35 GHz cloud radar plus a RWP has the specific advantage that it is only sensitive to particles and does not show any influence of Bragg scattering. The authors present a detailed methodology for the approach using Doppler spectra from the clear air and cloud returns. Overall I like the approach but I have some concerns about its robustness since some parameter adjustments are still required. The two cases presented with the separation approach are too limited to adequately assess the technique. The authors

should improve the meteorological discussion since they provide magnitudes of air motion and terminal velocity but they do not mention much whether the terminal velocity are realistic given the measured reflectivity in the same region. Other Comments: 1.) The results in the paper are intentionally focused on meteorological situations that are low rain rate, low terminal velocity, and low air motions. The Ka-band should be able to penetrate higher reflectivity clouds – the examples in the paper max out at around 10 dBZ. Can you say more about the upper limits of applicability of both the approach, and Ka-band radar? Does it matter if the Ka-band returns are attenuated in terms of the Cloud Radar/RWP spectra separation? William's use of two RWP allows for studies of more intense precipitation systems. 2.) Page 13, Line 31: I'm not clear how the dynamics are a key driver for this cloud. I see upward vertical motion at 5500 m at approximately 16:48 UTC. Is it possible that ice particles are formed aloft in this updraft region, and they fall down through the detected layer. The vertical motions below 5000 m are generally not upward and they do not seem to support ice particle growth. 3.) Page 12, lines 3-13: The LDR threshold of -13 dB based on Matrosov (1991) seems somewhat arbitrary. How good is your LDR calibration since this calibration can sometimes be challenging. What fraction of the data with detectable cloud reflectivity is deleted based on this threshold. 4.) The reference list is appropriate and the technical aspects of the paper in the paper are acceptable.
* * *

---

## Author Comment (AC1) · 17 Sep 2018

The comment was uploaded in the form of a supplement:
https://www.atmos-meas-tech-discuss.net/amt-2018-125/amt-2018-125-AC1-supplement.pdf

---

## Author Response (AR1)

**Reply to Referee #1**

Thank you for revising the manuscript and providing clear and constructive comments for improving it. Below you find our detailed response to each question.

Specific Comments

1. Equations (1), (5), (6), and (7). The term dˆ3 is not needed in these equations and is not defined in the text. I expect that "d" is raindrop diameter, but "D" is defined as the raindrop diameter in the text. Also, the variable eta(r) denoting the volume reflectivity includes the influence of raindrop diameter. Thus, raindrop diameter does not need to be explicitly stated in the equations.

In the context of this formula the term $d^3$ was intended to denote the differential in the three-dimensional volume integral over the resolution volume. D is used to clarify the dependence of backscattering cross section on particle size and number-size distribution.

We changed the description of Eq. 3 to: "where $\sigma(D)$ is the backscattering cross section **of particles with diameter D**" in order to make the meaning of this sentence more clear.

2. Page 3, line 10. For completeness, the backscattered cross section sigma(D) should be a function of radar operating frequency (or wavelength). Since the backscattering cross section at 35 GHz includes both Rayleigh scattering for small drops and non-Rayleigh scattering from larger drops, this should be reflected in the variable of sigma(D).

The subscript $\lambda$ was added in the formula and the text: "where $\sigma_\lambda(D)$ is the backscattering cross section of particles with diameter D **at a wavelength $\lambda$.**"

3. Page 3, line 10, equation (1) and also page 9, lines 1-14. Since the calibration is using small raindrops that are in the Rayleigh regime, it would make sense to include after equation (1) the Rayleigh and non-Rayleigh dependence in the 35 GHz backscattering cross-section (see comment #2 above). While this is discussed in the Discussion section (page 21, line 1), Rayelgih and non-Rayleigh scattering should be mentioned near the text developing equation (1).

Below equation 3 (in which the scattering cross section appears first) an explanatory text was added: **A simple analytical relationship for $\sigma_\lambda$ is available if the Rayleigh approximation holds, i.e. the particles are small compared to the wavelength. In the case of larger particles more complex approaches are required to relate $\sigma_\lambda$ to particle size and shape. In the following, signals originating from such particles is referred to as non-Rayleigh scattering.**

4. Page 9, lines 1-14. Is there any attenuation correction performed while implementing the calibration procedure? Even if there is not an attenuation correction performed, please include in the manuscript the need to account for attenuation at 35 GHz while performing the reflectivity estimation. Also, is this calibration procedure only performed at low altitudes, when the attenuation will be smaller than at further ranges?

When estimating the calibration constant, all profiles are excluded that show significant attenuation by liquid water. Hence, all profiles with reflectivity greater than 5 dBZ close to ground are neglected. The figure below shows the variation of calibration constant with height. Calibration is stable within a range of 2dB.

After first paragraph of section 4.2 we added: **Attenuation by gases is corrected by using the model of Liebe (1985, Radio Sci.). In this relative calibration scheme highly accurate absolute calibration of the cloud radar is not required, hence attenuation is not a primary concern.**

[Figure]

5. Page 4, lines 1-12, and Fig. 2. The diagram shown in Fig. 2 reminds me of a figure from Ralph (J. Atmos. Oceanic Technol. 257-267, 1995) showing the sensitivities of Bragg and Rayleigh scattering. But, the Ralph figure only goes up to a frequency of 10 GHz.

The figure of Ralph (1995, J. Atmos. Oceanic Technol.) shows reflectivity (dBZ) vs logarithm of Cn2. Our representation is more similar to the one shown by Gossard and Strauch (1983) with wavelength vs reflectivity. We have added a reference to both works in the figure caption.

6. Page 8, line 1. Please clarify the expression, "is calculated by subtracting the RWP retrieved air velocity from the cloud radar derived Doppler velocity". This "subtraction" could mean the subtraction of power within Doppler velocity spectrum, or could mean the subtraction of velocity moments between different radars. Also, the "subtraction" could mean the shift in Doppler velocity spectrum before estimating the moments. I thought the rest of the manuscript read very well with good descriptions of the algorithm and the applications.

We have refined the sentence: The particle terminal fall velocity is calculated by subtracting the **vertical air velocity (first moment of the particle suppressed RWP spectrum) from the cloud radar vertical velocity.**

**Reply to Referee #2**

Thank you for carefully reading the manuscript and providing clear and constructive comments for improving the manuscript. Below each question is addressed in detail.

The paper presents a method to separate cloud/precipitation radar returns from clear air returns using a 35 GHz cloud radar and a 482 MHz radar wind profiler (RWP). Previous work by Williams and others has shown the utility of using two RWP (UHF and VHF) to separate air motions and hydrometeor terminal velocity. The use of a 35 GHz cloud radar plus a RWP has the specific advantage that it is only sensitive to particles and does not show any influence of Bragg scattering. The authors present a detailed methodology for the approach using Doppler spectra from the clear air and cloud returns. Overall I like the approach but I have some concerns about its robustness since some parameter adjustments are still required. The two cases presented with the separation approach are too limited to adequately assess the technique.

In the present paper we wanted to focus on the technical details of the algorithm. We acknowledge that a comprehensive verification is still pending and would require to incorporate a longer dataset. This is beyond the scope of this work and is intended to be subject of follow up papers.

The authors should improve the meteorological discussion since they provide magnitudes of air motion and terminal velocity but they do not mention much whether the terminal velocity are realistic given the measured reflectivity in the same region.

We acknowledge that an independent validation of the presented terminal velocities is missing. Nevertheless, we think that the available Z-vt relationships will serve not serve purpose. Firstly, they depend crucially on absolute calibration and attenuation correction of the cloud radar signal. Our approach only relies on the relative calibration between RWP and cloud radar and extensive treatment of cloud radar and, as mentioned before, absolute calibration is beyond the scope of this study. Secondly, magnitude of the vertical motion is below 0.4 m/s (see e.g. Fig. 9). Comparing with established Z-vt relationships our velocities are within the internal variability of those relationships.

Other Comments:

1.) The results in the paper are intentionally focused on meteorological situations that are low rain rate, low terminal velocity, and low air motions. The Ka-band should be able to penetrate higher reflectivity clouds – the examples in the paper max out at around 10 dBZ. Can you say more about the upper limits of applicability of both the approach, and Ka-band radar? Does it matter if the Ka-band returns are attenuated in terms of the Cloud Radar/RWP spectra separation? William's use of two RWP allows for studies of more intense precipitation systems.

We agree that the 35 GHz Radar is limited in its capability to penetrate strong precipitation and that a combination of two RWPs is better suited for investigating intense precipitation. The applicability of the algorithm is limited in cases of strong attenuation at the cloud radar frequency. Weak attenuation can nevertheless be corrected as long as the relative shape of the spectrum is preserved . We assume that this is the case for rain events with a precipitation intensity below 10mm/h, but a more solid statement would require a longer dataset as those events are rather rare at Lindenberg (below 0.1% of precipitation time). Nevertheless, we are focusing more on cloud microphysics studies under stratiform conditions. Attenuation from intense precipitation is, hence, not a major concern of this study.

2.) Page 13, Line 31: I'm not clear how the dynamics are a key driver for this cloud. I see upward vertical motion at 5500 m at approximately 16:48 UTC. Is it possible that ice particles are formed aloft in this updraft region, and they fall down through the detected layer. The vertical motions below 5000 m are generally not upward and they do not seem to support ice particle growth.

We agree that the statement is ambiguous. The microphysical processes within such a cloud layer are mostly driven by turbulence at cloud top, aerosol load and ice formation within the cloud top layer. A large updraft as it is visible in Fig. 10, however, can influence the cloud in two significant ways: Firstly, an updraft like this could be responsible for aerosol activation through adiabatic supersaturation and, hence, for actual "creation" of such a cloud layer. Secondly, up- and downdrafts on the scale-length of the cloud can also enhance or decrease the growth of ice particles inside of the cloud. In this way, vertical motions are critical for the life cycle of the cloud. Seeding from aloft is also unlikely, as the layer above the cloud is very dry (spread > 8 K; Fig 10 a) and no cirrus clouds could be observed aloft (not shown).

We have removed the word "key" in the text: As the vertical air motion reveals in Fig. 11 b, **the cloud exists under a background of strong vertical motions.**

3.) Page 12, lines 3-13: The LDR threshold of -13 dB based on Matrosov (1991) seems somewhat arbitrary. How good is your LDR calibration since this calibration can sometimes be challenging. What fraction of the data with detectable cloud reflectivity is deleted based on this threshold.

The calibration was done following Myagkov et al (2015, J. Atmos. Oceanic Technol.) with an integrated cross-polarization ratio (ICPR) of -25 dB. The uncertainty is assumed to be below 3 dB, especially as the impact decreases for high values of LDR (i.e. for an observed LDR of -15.00 dB the corrected value would be -15.45 dB).

The threshold was selected to exclude return from non-hydrometeor scattering, like insects and pollen. The simulations of Matrosov (1991, J. Atmos. Sci.) showed that hydrometors can only cause LDRs below -15 dB , a threshold which was already used in several studies (e.g. Illingworth 2007, Bull. Am. Meteorol. Soc.; Bühl 2016, Atmos. Chem. Phys.). Our choice of -13 dB is rather conservative.

The plankton filtering is done in two steps which were not clearly separated in the original version of the manuscript. Firstly, the cloud radar spectrum is filtered for bins containing high LDR to prohibit a misinterpretation as hydrometeors during the separation. After the separation, the LDR is used to flag spectra, where the reflectivity caused by plankton was higher than the RWP noise level (e.i. could potentially be detected by the RWP). Nevertheless, in the atmospheric boundary layer – where the plankton occurs – the reflectivity of Bragg return observed by the RWP is at least 10-15 dB higher than the plankton signal.

In section 4.1 we added:

**Before the broadening, bins in the cloud radar that are dominated by plankton (like insects or pollen) are removed by using the linear depolarization ratio (LDR) to identify highly depolarizing targets (Reinking 1997). All bins with a LDR higher than -13 dB are filtered, because hydrometeors exhibit a smaller LDR (e.g. Matrosov 1991).**

And in section 4.4 we modified the part on plankton to:

**As stated in section 4.1 a LDR threshold of -13 dB is used to separate scattering from hydrometeors from atmospheric plankton. The plankton flag is set, when the LDR of the whole cloud radar spectrum is above the LDR threshold and the total reflectivity is low, but above the RWP noise level. For scattering by plankton the cloud radar frequently shows the same vertical air velocity as the RWP.**

4.) The reference list is appropriate and the technical aspects of the paper in the paper are acceptable.

**Minor Changes**

[revised manuscript text omitted]